# NAS-BENCH-SUITE:
# NAS EVALUATION IS (NOW) SURPRISINGLY EASY

**Yash Mehta**[1,*] **Colin White**[2,*] **Arber Zela**[1], **Arjun Krishnakumar**[1],
**Guri Zabergja**[1], **Shakiba Moradian**[1], **Mahmoud Safari**[1], **Kaicheng Yu**[2], **Frank Hutter**[1,3]
[1] University of Freiburg, [2] Abacus.AI, [3] Bosch Center for AI

## ABSTRACT

The release of tabular benchmarks, such as NAS-Bench-101 and NAS-Bench-201, has significantly lowered the computational overhead for conducting scientific research in neural architecture search (NAS). Although they have been widely adopted and used to tune real-world NAS algorithms, these benchmarks are limited to small search spaces and focus solely on image classification. Recently, several new NAS benchmarks have been introduced that cover significantly larger search spaces over a wide range of tasks, including object detection, speech recognition, and natural language processing. However, substantial differences among these NAS benchmarks have so far prevented their widespread adoption, limiting researchers to using just a few benchmarks. In this work, we present an in-depth analysis of popular NAS algorithms and performance prediction methods across 25 different combinations of search spaces and datasets, finding that many conclusions drawn from a few NAS benchmarks do *not* generalize to other benchmarks. To help remedy this problem, we introduce `NAS-Bench-Suite`, a comprehensive and extensible collection of NAS benchmarks, accessible through a unified interface, created with the aim to facilitate reproducible, generalizable, and rapid NAS research. Our code is available at `https://github.com/automl/naslib`.

## 1 INTRODUCTION

Automated methods for neural network design, referred to as neural architecture search (NAS), have been used to find architectures that are more efficient and more accurate than the best manually designed architectures (Zoph et al., 2018; Real et al., 2019; So et al., 2019). However, it is notoriously challenging to provide fair comparisons among NAS methods due to potentially high computational complexity (Zoph & Le, 2017; Real et al., 2019) and the use of different training pipelines and search spaces (Li & Talwalkar, 2019; Lindauer & Hutter, 2020), resulting in the conclusion that *"NAS evaluation is frustratingly hard"* (Yang et al., 2020). To make fair, statistically sound comparisons of NAS methods more accessible, tabular NAS benchmarks have been released; these exhaustively evaluate all architectures in a given search space, storing the relevant training metrics in a lookup table (Ying et al., 2019; Dong & Yang, 2020; Zela et al., 2020b; Mehrotra et al., 2021). This substantially lowers the computational overhead of NAS experiments, since the performance of an architecture can be found simply by querying these tables, hence allowing for a rigorous comparison of various NAS algorithms with minimal computation.

While early tabular NAS benchmarks, such as NAS-Bench-101 (Ying et al., 2019) and NAS-Bench-201 (Dong & Yang, 2020), have been widely adopted by the community, they are limited to small search spaces and focus solely on image classification. Recently, benchmarks have been introduced for natural language processing (Klyuchnikov et al., 2020), speech recognition (Mehrotra et al., 2021), object detection, and self-supervised tasks (Duan et al., 2021). Furthermore, the release of *surrogate* NAS benchmarks (Siems et al., 2020; Yan et al., 2021), which estimate the performance of all architectures in a search space via a surrogate model, has removed the constraint of exhaustively evaluating the entire search space, expanding the scope of possible search space sizes to $10^{18}$ and beyond. However, substantial differences in the abstractions (such as whether a node or an edge

---

*Equal contribution. Email to: `{mehtay,zelaa}@cs.uni-freiburg.de, colin@abacus.ai`

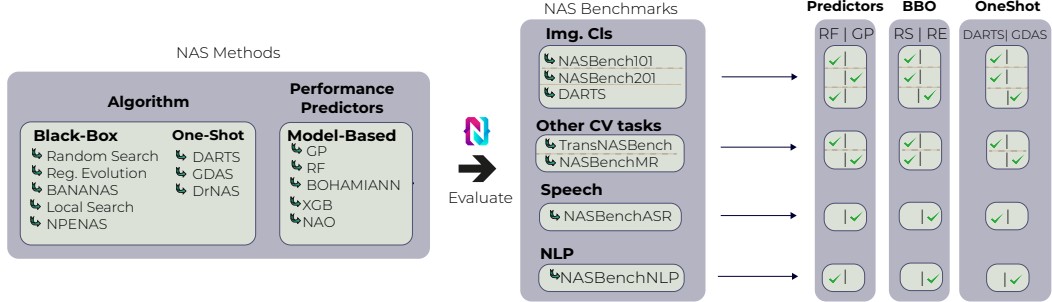

Figure 1: Overview of `NAS-Bench-Suite`.

denotes an operation), capabilities (such as whether all, or only some, of the architectures can be queried), and implementations (such as incompatible deep learning libraries) have so far prevented nearly all research in NAS from providing results on more than two families of benchmarks. Overall, the lack of consistency in "NAS-Bench" datasets has significantly slowed their collective adoption.

In this work, we show that there is a need to adopt newer benchmarks because many conclusions drawn from a small subset of benchmarks do not generalize across diverse datasets and tasks. Specifically, we present an in-depth analysis of popular black-box (Real et al., 2019; White et al., 2021a; Ottelander et al., 2021), one-shot (Liu et al., 2019b; Chen et al., 2021; Dong & Yang, 2019), and performance prediction methods (White et al., 2021c) across (nearly) every publicly available queryable NAS benchmark. This includes 25 different combinations of search spaces and datasets, which is, to the best of our knowledge, by far the largest set of NAS search spaces and datasets on which experiments have been conducted to date. We show that many implicit assumptions in the NAS community are wrong. First, if a NAS algorithm does well on NAS-Bench-101 and NAS-Bench-201, it does not necessarily perform well on other search spaces. Second, NAS algorithms may not have robust default hyperparameters and therefore require tuning. Finally, tuning the hyperparameters of a NAS method on one search space and transferring these hyperparameters to other search spaces often make the NAS method perform significantly worse.

In order to help NAS researchers and practitioners avoid these pitfalls, we release the NAS Benchmark Suite (`NAS-Bench-Suite`), a comprehensive and extensible collection of NAS benchmarks, accessible through a unified interface, created with the aim to facilitate reproducible, generalizable, and rapid NAS research. Our work eliminates the overhead for NAS research to evaluate on several different datasets and problem types, helping the community to develop NAS methods that generalize to new problem types and unseen datasets. See Figure 1 for an overview. To ensure reproducibility and other best practices, we release our code and adhere to the NAS best practices checklist (Lindauer & Hutter, 2020, see Section A for details).

**Our contributions.** We summarize our main contributions below.

- We conduct a comprehensive study of the generalizability of NAS algorithms and their hyperparameters across 25 settings, showing that it is often not sufficient to tune on just a few benchmarks, and showing that the best hyperparameters depend on the specific search space.
- We introduce a unified benchmark suite, `NAS-Bench-Suite`, which implements nearly every publicly available queryable NAS benchmark – 25 different combinations of search spaces and datasets. By making it easy to quickly and comprehensively evaluate new NAS algorithms, our benchmark suite can improve experimental rigor and generalizability in NAS research.

## 2 NAS BENCHMARKS OVERVIEW

**Preliminaries.** A *search space* in NAS is the set of all architectures that the NAS algorithm is allowed to select. Most recent search spaces are defined by a *cell-based (micro)* structure and a *macro* structure. A *cell* is a small set of neural network operations arranged in a directed acyclic graph (DAG), with constraints on the number of nodes, edges, and incoming edges per node. The *macro* structure consists of the architecture skeleton and the arrangement of cells, such as how many times each cell is duplicated. For many popular search spaces, the macro structure is completely fixed,

Table 1: Overview of NAS benchmarks in `NAS-Bench-Suite`.

| Benchmark | Size | Queryable Tab. | Queryable Surr. | LCs | Macro | Type | #Tasks | NAS-Bench-Suite |
|---|---|---|---|---|---|---|---|---|
| NAS-Bench-101 | 423k | ✓ | | | | Image class. | 1 | ✓ |
| NAS-Bench-201 | 6k | ✓ | | ✓ | | Image class. | 3 | ✓ |
| NAS-Bench-NLP | $10^{53}$ | | | ✓ | | NLP | 1 | ✓ |
| NAS-Bench-1Shot1 | 364k | ✓ | | | | Image class. | 1 | ✓ |
| NAS-Bench-301 | $10^{18}$ | | ✓ | | | Image class. | 1 | ✓ |
| NAS-Bench-ASR | 8k | ✓ | | | ✓ | ASR | 1 | ✓ |
| NAS-Bench-MR | $10^{23}$ | | ✓ | | ✓ | Var. CV | 4 | ✓ |
| TransNAS-Bench | 7k | ✓ | | ✓ | ✓ | Var. CV | 14 | ✓ |
| NAS-Bench-111 | 423k | | ✓ | ✓ | | Image class. | 1 | ✓ |
| NAS-Bench-311 | $10^{18}$ | | ✓ | ✓ | | Image class. | 1 | ✓ |
| NAS-Bench-NLP11 | $10^{53}$ | | ✓ | ✓ | | NLP | 1 | ✓ |

while for other search spaces, the macro structure can have variable length, width, and number of channels for different architectures in the search space.

A *NAS benchmark* (Lindauer & Hutter, 2020) consists of a dataset (with a fixed train-test split), a search space, and a fixed evaluation pipeline with predefined hyperparameters for training the architectures. A *tabular* NAS benchmark is one that additionally provides precomputed evaluations with that training pipeline for all possible architectures in the search space. Finally, a *surrogate* NAS benchmark (Siems et al., 2020; Yan et al., 2021) is a NAS benchmark that provides a surrogate model that can be used to predict the performance of any architecture in the search space. We say that a NAS benchmark is *queryable* if it is either a tabular or surrogate benchmark. Queryable NAS benchmarks can be used to simulate NAS experiments very cheaply by querying the performance of neural networks (using a table or a surrogate) instead of training the neural networks from scratch.

**NAS benchmarks.** Now we describe the characteristics of many popular NAS benchmarks. For a summary, see Table 1, and for a more comprehensive and detailed survey, see Appendix B. The first tabular NAS benchmark to be released was NAS-Bench-101 (Ying et al., 2019). This benchmark consists of $423\,624$ architectures trained on CIFAR-10. The cell-based search space consists of a directed acyclic graph (DAG) structure in which the nodes can take on operations. A follow-up work, NAS-Bench-1Shot1 (Zela et al., 2020b), defines three subsets of NAS-Bench-101 which allow one-shot algorithms to be run. The largest subset size in NAS-Bench-1Shot1 is $363\,648$. NAS-Bench-201 Dong & Yang (2020) is another popular tabular NAS benchmark. The cell-based search space consists of a DAG where each *edge* can take on operations (in contrast to NAS-Bench-101, in which the *nodes* are operations). The number of non-isomorphic architectures is $6\,466$ and all are trained on CIFAR-10, CIFAR-100, and ImageNet-16-120. NATS-Bench (Dong et al., 2021) is an extension of NAS-Bench-201 which also varies the macro architecture.

NAS-Bench-NLP (Klyuchnikov et al., 2020) is a NAS benchmark for natural language processing, which is size $10^{53}$. However, only $14\,322$ of the architectures were trained on Penn Tree-Bank (Mikolov et al., 2010), meaning NAS-Bench-NLP is not queryable. NAS-Bench-ASR (Mehrotra et al., 2021) is a tabular NAS benchmark for automatic speech recognition. The search space consists of $8\,242$ architectures trained on the TIMIT dataset. TransNAS-Bench (Duan et al., 2021) is a tabular NAS benchmark consisting of two separate search spaces (cell-level and macro-level) and seven tasks including pixel-level prediction, regression, and self-supervised tasks. The cell and macro search spaces are size $4\,096$ and $3\,256$, respectively. NAS-Bench-MR (Ding et al., 2021) is a surrogate NAS benchmark which evaluates across four datasets: ImageNet50-1000, Cityscapes, KITTI, and HMDB51. NAS-Bench-MR consists of a single search space of size $10^{23}$.

The DARTS (Liu et al., 2019b) search space with CIFAR-10, consisting of $10^{18}$ architectures, is arguably the most widely-used NAS benchmark. Recently, $60\,000$ of the architectures were trained

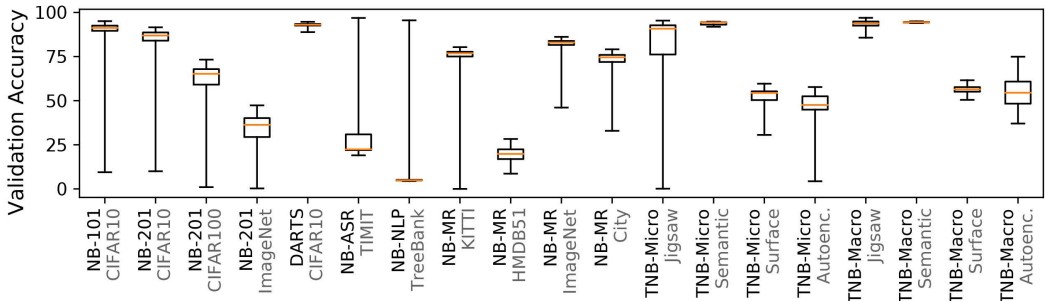

Figure 2: Validation accuracy box plots for each NAS benchmark. The whiskers represent the minimum and maximum accuracies in each search space. For NAS-Bench-NLP and TransNAS-Bench, perplexity and SSIM are used instead of validation accuracy, respectively. In the case of extremely large search spaces such as DARTS and NAS-Bench-NLP, the statistics are computed only with respect to the tens-of-thousands of precomputed architectures.

and used to create NASBench-301 (Siems et al., 2020), the first surrogate NAS benchmark. More recently, NAS-Bench-111, NAS-Bench-311, and NAS-Bench-NLP11 (Yan et al., 2021) were released as surrogate benchmarks that extend existing benchmarks with the full learning curve information.

## 3 NAS BENCHMARK STATISTICS

In this section and in Appendix C, we use `NAS-Bench-Suite` to compute a set of aggregate statistics across a large set of NAS benchmarks. There is a high variance with respect to the distribution of accuracies and other statistics across benchmarks due to substantial differences in the tasks performed and layout of the search space. It is essential to keep this in mind to ensure a fair comparison of the performance of NAS algorithms across these benchmarks. To the best of our knowledge, this is the first large-scale aggregation of statistics computed on NAS benchmarks.

Figure 2 shows box plots for the validation accuracy distribution for a representative set of the 25 NAS benchmarks. We find that TransNAS-Bench (Sem. Segment) and DARTS achieve the highest median and maximum accuracies, yet they also have among the smallest variance in validation accuracy across the search space. On the other hand, the search space with the highest interquartile range is TransNAS-Bench Jigsaw. In Figure 3, we assess the level of *locality* in each search space, or the similarity of validation accuracy among neighboring architectures (architectures which differ by a single operation or edge) using the random walk autocorrelation (RWA) (Weinberger, 1990; Ying et al., 2019; White et al., 2021b). RWA computes the autocorrelation of accuracies of architectures during a random walk, in which each step perturbs one operation or edge. We see that NAS-Bench-201 ImageNet16-120 has the highest autocorrelation, while NAS-Bench-101 has the lowest. In Appendix C, we also discuss plots describing the average runtime for training architectures and the average neighborhood size, for each NAS benchmark. Overall, we see substantial differences among the search spaces along the various axes that we tested.

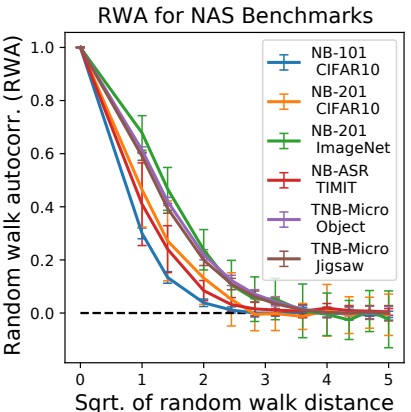

Figure 3: RWA for NAS benchmarks. RWA computes the autocorrelation of accuracies of architectures during a random walk, in which each step perturbs one operation or edge.

Overall, we find that the diversity is important to keep into context when comparing across many different NAS benchmarks. For example, it is more impressive if a NAS algorithm discovers an architecture within $0.1\%$ of the optimal on NAS-Bench-201 ImageNet16-120, compared to DARTS, because the standard deviation of accuracies for DARTS is much lower. Additional factors affect the

difficulty of NAS benchmarks for some NAS algorithms more than for others; for example, locality has a large effect on the performance of regularized evolution but not for random search.

# 4 ON THE GENERALIZABILITY OF NAS ALGORITHMS

In this section, we carry out a large-scale empirical study on the generalizability of NAS algorithms across diverse search spaces and tasks, using five different black-box algorithms, five different performance predictors, and three one-shot methods across the largest set of NAS benchmarks to date. Throughout, we empirically assess three assumptions we have witnessed in the NAS community about the generalizability of NAS algorithms across diverse search spaces and tasks:

1. "If a NAS algorithm does well on the popular NAS benchmarks NAS-Bench-101 and all three datasets of NAS-Bench-201, it surely must generalize to other NAS benchmarks."
2. "NAS algorithms tend to have robust default hyperparameters and do not require tuning."
3. "To improve a NAS algorithm on a new benchmark, we can cheaply optimize its hyperparameters on a tabular benchmark and then transfer the optimized hyperparameters."

**Experimental details.** A black-box NAS algorithm is an algorithm which iteratively chooses architectures to train, and then uses the final validation accuracies in the next iteration. We run experiments for five popular black-box NAS algorithms: random search (RS) (Li & Talwalkar, 2019; Sciuto et al., 2020), regularized evolution (RE) (Real et al., 2019), local search (LS) (White et al., 2021b; Ottelander et al., 2021), BANANAS (White et al., 2021a), and NPENAS (Wei et al., 2020). We run each black-box algorithm for 200 iterations.

Recently, model-based performance prediction methods have gained popularity as subroutines to speed up NAS algorithms (Ning et al., 2020). These methods work by training a model using a set of already evaluated architectures, and then using the model to predict the performance of untrained architectures. We compare five popular performance predictors: BOHAMIANN (Springenberg et al., 2016), Gaussian process (GP) (Rasmussen, 2003), random forest (RF) (Breiman, 2001), neural architecture optimization (NAO) (Luo et al., 2018), and XGBoost (Chen & Guestrin, 2016). We evaluate the performance prediction methods by computing the Spearman rank correlation of the predictions versus ground truth validation accuracy on a held-out test set of 200 architectures. For each black-box method and each performance predictor, we evaluate the default hyperparameter configuration, as well as 300 randomly sampled hyperparameter configurations, with each reported performance averaged over 10 seeds. We give additional details of each method in Appendix C.

## 4.1 THE BEST NAS METHODS

In Figure 4, we plot the scaled (relative) performance of all five black-box algorithms and performance predictors across a representative set of NAS benchmarks (with the full plot in Appendix C). In Table 2, we give a summary by computing the average rank of each black-box algorithm or performance prediction method across all 25 NAS benchmarks.

Across black-box algorithms with their default hyperparameters, we find that no algorithm performs well across all search spaces: no algorithm achieves an average rank close to 1 across all search spaces. RE and LS perform the best on average across the search spaces, with average rankings of 2.36 and 2.66, respectively. We also find that although RE performed the best on average, it performs worse than random search in three cases. Therefore, there is no "best" black-box algorithm. When comparing black-box algorithms tuned on each individual benchmark, RE achieves a ranking of 1.96, although we note that since black-box algorithms are expensive to evaluate, it is computationally prohibitive to tune for each individual NAS benchmark.

Across performance predictors, we find that the best predictor with default parameters is RF, and the best predictor when tuned on each individual benchmark is XGBoost, with average rankings of 1.57 and 1.23, respectively. Note that since performance prediction subroutines are often not the bottleneck of NAS, it is common to run hyperparameter tuning during the NAS search. Therefore, we conclude that XGBoost (when tuned) *does* generalize well across all 25 search spaces we tested.

**Generalizing beyond NAS-Bench-101 and -201.** Now we test how well NAS methods generalize from NAS-Bench-101 and -201 to the rest of the NAS benchmarks. In Table 2, for both black-box

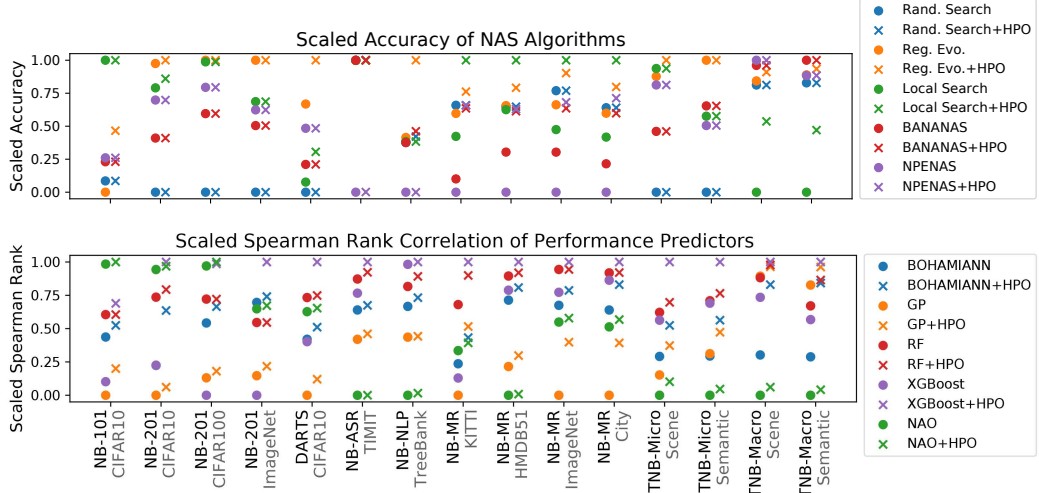

Figure 4: Relative performance of black-box algorithms (top) and performance predictors (bottom) across NAS benchmarks. The solid circles show the performance of the algorithm with default hyperparameters, while the crosses show performance after hyperparameter optimization (HPO).

and predictor methods, we compare the average rank of each method across two different subsets of benchmarks: NAS-Bench-101 and the three different datasets of NAS-Bench-201, versus the rest of the 21 settings excluding NAS-Bench-101 and NAS-Bench-201. We find that for both black-box method and performance predictor methods, the best method *substantially* changes between these two subsets. For example, NAO is the top-performing predictor across NAS-Bench-101 and NAS-Bench-201, yet it achieves very poor performance on the rest of the benchmarks. This suggests that the insights derived from empirical results are highly dependent on the benchmarks used, and that in order to make reliable claims, evaluating on more than a few benchmarks is crucial.

Table 2: Average relative performance ranking among five NAS algorithms (left) or five performance predictors (right) across 25 settings. Results are weighted by search space; e.g., each of the three NAS-Bench-201 benchmarks are weighted by 1/3. For abbreviations, see Table 3.

| | | NAS Algorithms | | | | | Performance Predictors | | | |
| | RS | RE | BANANAS | LS | NPENAS | GP | BOHAM. | RF | XGB | NAO |
|---|---|---|---|---|---|---|---|---|---|---|
| Avg. Rank | 3.47 | **2.36** | 3.02 | 2.66 | 3.48 | 4.25 | 3.00 | **1.57** | 2.95 | 3.25 |
| Avg. Rank, HPO | 3.97 | **1.96** | 3.17 | 2.41 | 3.49 | 4.37 | 3.36 | 2.41 | **1.23** | 3.62 |
| Avg.Rank, 101&201 | 4.50 | 3.00 | 3.50 | **1.50** | 2.50 | 4.67 | 2.83 | 2.17 | 4.17 | **1.17** |
| Avg. Rank, non-101&201 | 3.06 | **2.11** | 2.83 | 3.13 | 3.87 | 4.08 | 3.06 | **1.33** | 2.46 | 4.08 |

## 4.2 GENERALIZABILITY OF HYPERPARAMETERS

While the previous section assessed the generalizability of NAS methods, now we assess the generalizability of the *hyperparameters* within NAS methods. For a given NAS method, we can tune it on NAS benchmark $A$, and then evaluate the performance of the tuned method on NAS benchmark $B$, compared to the performance of the best hyperparameters from NAS benchmark $B$. In other words, we compute the "regret" of tuning a method on one NAS benchmark and deploying it on another.

In Figure 5 (left), we run this experiment for all pairs of search spaces, averaged over all performance predictors, to give a general estimate of the regret across all search spaces. Unsurprisingly, hyperparameters transfer well within a given search space (such as within the three datasets in NAS-Bench-201 or the seven datasets in TransNAS-Bench-Micro). However, we find that no search space achieves strong regret across most search spaces. NAS-Bench-101, DARTS, and the four benchmarks in NAS-Bench-MR have particularly bad regret compared to the other benchmarks.

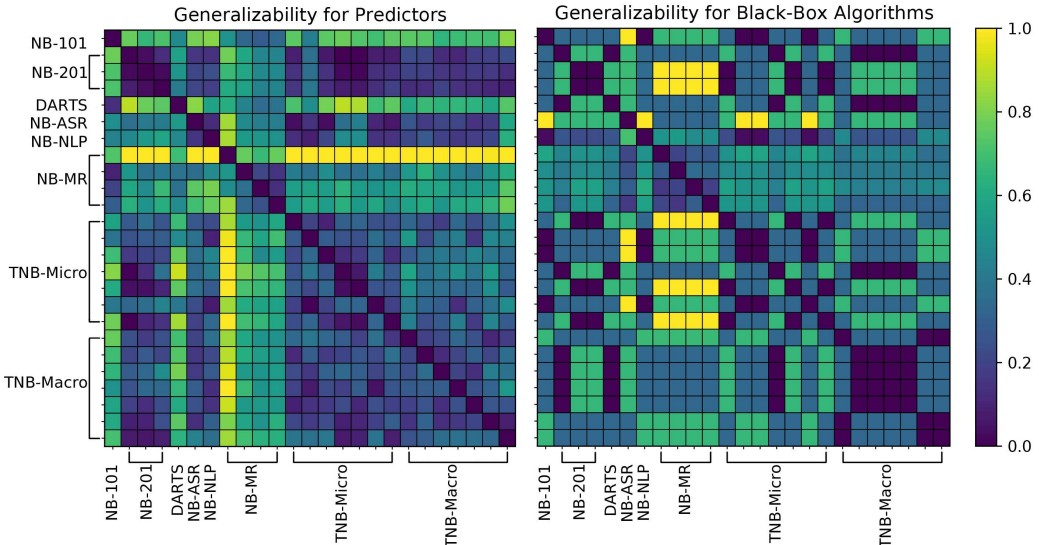

Figure 5: Transferability results for predictors (left) and black-box algorithms (right). Row $i$, column $j$ denotes the scaled regret of an algorithm tuned on search space $i$ and evaluated on search space $j$. For abbreviations, see Table 3, and for summary statistics, see Appendix D.2.

Next, in Figure 5 (right), we run the same experiment for black-box algorithms. We find that the transferability of hyperparameters across black-box algorithms is even worse than across predictors: the hyperparameters do not always transfer well even within different tasks of a fixed search space. We also see that, interestingly, the matrix is less symmetric than for performance predictors. For example, it is particularly hard for hyperparameters to transfer *to* NAS-Bench-MR, but easier to transfer *from* NAS-Bench-MR. Overall, our experiments show that it is not sufficient to tune hyperparameters on one NAS benchmark and deploy on other benchmarks, as this can often make the performance worse. In Appendix D, we give additional experiments and summary statistics on the transferability of hyperparameters across search spaces, as well as a guide to interpreting the experiments. We also present additional experiments that combine our algorithm and statistics experiments to give relationships between properties of the search space and the performance of different algorithms.

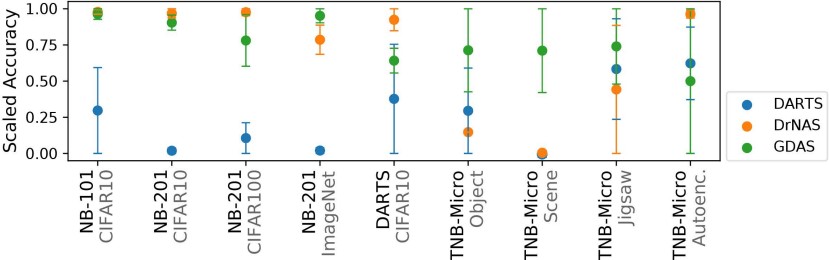

Figure 6: Performance of one-shot algorithms across NAS benchmarks. The bars show the minimum, maximum and average performance over five seeds. For abbreviations, see Table 3.

## 4.3 ONE-SHOT ALGORITHMS

One-shot NAS algorithms, in which a single supernetwork representing the entire search space is trained, are a popular choice for NAS due to their strong performance and fast runtimes. In this section, we compare the performance of three one-shot algorithms: DARTS (Liu et al., 2019b), GDAS (Dong & Yang, 2019), and DrNAS (Chen et al., 2021), across several different NAS benchmarks. Note that since one-shot algorithms must be able to represent the entire search space in the form of a supernetwork, the algorithms can effectively only be run on cell-based search spaces with a complete graph (Zela et al., 2020b), precluding the use of all 25 NAS benchmarks as in the previous section.

In Figure 6, we plot the scaled (relative) performance of the three one-shot methods run for five seeds each, across nine NAS benchmarks. There is no clear best algorithm: DrNAS performed best on five benchmarks, and GDAS performed best on the remaining four. DARTS did not perform as well, which is consistent with prior work (Zela et al., 2020a; Dong & Yang, 2020).

Throughout this section, we showed that many implicit assumptions in the NAS community regarding NAS algorithm generalizability are incorrect, and that it is important to consider a large set of NAS benchmarks in order to avoid false conclusions. In the next section, in order to help NAS researchers and practitioners avoid these pitfalls, we describe our new NAS benchmark suite, which is designed with the aim to help the community develop generalizable NAS methods.

## 5  NAS-BENCH-SUITE: OVERVIEW

In this section, we give an overview of the `NAS-Bench-Suite` codebase, which allows researchers to easily compare NAS algorithms across many tasks, as shown in Section 4.

### 5.1  KEY CHALLENGES IN CREATING A FLEXIBLE BENCHMARK SUITE

Before the introduction of tabular NAS benchmarks, it was common practice to release new NAS methods along with custom-designed search spaces to go with the methods (Lindauer & Hutter, 2020; Li & Talwalkar, 2019), leading to many early NAS methods being intertwined with and hard to separate from their original search space. The result was that many algorithms needed substantial code changes in order to run on other search spaces.

Even after the release of several queryable benchmarks, it is still not common practice to run NAS algorithms on more than a few benchmarks due to the nontrivial differences among each benchmark. For example, as described in Section 2, operations on the nodes (Ying et al., 2019) versus on the edges (Liu et al., 2019b) added complexity in adapting one-shot optimizers to many search spaces, and for some search spaces one-shot optimizers could only be run on subsets of the full space (Zela et al., 2020b). Other differences, such as the presence of hidden nodes (Klyuchnikov et al., 2020) or skip connections (Mehrotra et al., 2021), cause NAS components

```
from naslib.search_spaces import NasBench101SearchSpace    1
from naslib.optimizers import RegularizedEvolution          2
from naslib.defaults.trainer import Trainer                 3
from naslib.utils import utils, get_dataset_api             4
                                                            5
config = utils.get_config_from_args(config_type='nas')      6
                                                            7
search_space = NasBench101SearchSpace()                     8
optimizer = RegularizedEvolution(config)                    9
                                                           10
dataset_api = get_dataset_api(config.search_space,         11
                              config.dataset)               12
                                                           13
optimizer.adapt_search_space(search_space,                 14
                             dataset_api=dataset_api)       15
trainer = Trainer(optimizer, config)                       16
trainer.search()                                           17
trainer.evaluate()                                         18
```

Snippet 1: A minimal example on how one can run a NAS algorithm in `NAS-Bench-Suite`. Both the search space and the algorithm can be changed in one line of code.

to require different implementations. Creating a robust NAS benchmark suite is not as simple as "combining the individual codebases", because such a solution would require re-implementing each new NAS algorithm on several search spaces. In fact, the difficulty in creating a benchmark suite is likely a key reason why there are so few papers that evaluate on more than a few benchmarks.

### 5.2  THE NAS-BENCH-SUITE CODEBASE

To overcome these difficulties, `NAS-Bench-Suite` enacts two key principles: flexibility in defining the search space and modularity of individual NAS components. We first describe how we achieve flexible search space definitions, and then we detail the modular design of `NAS-Bench-Suite`.

A search space is defined with a graph object using PyTorch and NetworkX (Hagberg et al., 2008), a package that allows for easy-to-use and flexible graph creation and manipulations. This functionality allows for a dynamic search space definition, where candidate operations can be encoded both as part of a node  (Ying et al., 2019; Klyuchnikov et al., 2020) as well as a edge  (Dong & Yang, 2020; Liu et al., 2019b). It also allows the representation of multiple layers of graphs on top of the computational graph, allowing the formation of nested graph-structures that can be used to define hierarchical spaces  (Ru et al., 2020; Liu et al., 2019a).

`NAS-Bench-Suite` is modular in the sense that individual NAS components, such as the search space or NAS algorithm, are disentangled and defined separately from one another. In Snippet 1, we showcase a minimal example that runs a black-box NAS algorithm on a tabular benchmark. Other optimizers and benchmarks can be imported and run similarly. Due to these design principles, `NAS-Bench-Suite` allows researchers to implement their NAS algorithm in isolation, and then evaluate on all the benchmarks integrated in `NAS-Bench-Suite` without writing any additional code. Since a variety of NAS algorithms, search spaces, and performance predictors have already been integrated into the open-source framework, this allows for the user to build on top of predefined NAS components. With the entire pipeline in place, along with the possibility of quick evaluations across search spaces and tasks, we believe that `NAS-Bench-Suite` will allow researchers to rapidly prototype and fairly evaluate NAS methods. All the scripts to run the evaluations conducted in this paper come together with the library codebase. For more details, see Appendix E.

## 6 RELATED WORK

We describe work that provides experimental surveys, benchmark suites, or unified codebases within NAS. For detailed surveys on NAS, see (Elsken et al., 2019; Xie et al., 2020).

**NAS experimental surveys.** Multiple papers have found that random search is a competitive NAS baseline (Li & Talwalkar, 2019; Sciuto et al., 2020), including a recent work that benchmarked eight NAS methods with five datasets on the DARTS search space (Yang et al., 2020). Other recent works have shown experimental surveys for NAS performance predictors (Ning et al., 2020; White et al., 2021c), and experimental analyses on weight sharing (Yu et al., 2020).

**Benchmark suites.** NAS-Bench-360 (Tu et al., 2021) is a very recent benchmark suite which presents NAS benchmarks for ten diverse datasets on three search spaces. However, a drawback is that evaluating NAS algorithms requires 1 to 100+ GPU-hours (Tu et al., 2021). This is in contrast to the `NAS-Bench-Suite`, where NAS algorithms take at most 5 minutes on a CPU due to the use of queryable benchmarks. Outside of NAS, multiple hyperparameter tuning benchmark suites have been released (Eggensperger et al., 2021; Arango et al., 2021).

**NAS codebases.** The *DeepArchitect* library (Negrinho & Gordon, 2017) was the first to have a modular design for search spaces and NAS algorithms. *PyGlove* (Peng et al., 2021) is a library for NAS featuring dynamically adapting components, however, it is not open-sourced. *Neural Network Intelligence* (NNI) (Microsoft, 2017) is a platform for AutoML that implements many algorithms as well as NAS-Bench-101 and NAS-Bench-201. Other NAS repositories are actively being built, such as *archai* (Shah & Dey, 2020) and *aw_nas* (Ning et al., 2020).

## 7 CONCLUSION AND FUTURE WORK

In a large-scale study across 25 NAS benchmark settings, 5 blackbox NAS methods, 5 NAS predictors, and 3 one-shot methods, we showed that many implicit assumptions in the NAS community are wrong. Firstly, there is no single best NAS method: which method performs best very much depends on the benchmark. Along similar lines, if a NAS method performs well on the popular NAS benchmarks NAS-Bench-101 and all three datasets of NAS-Bench-201, in contrast to what one might have expected, this does *not* imply that it will also perform well on other NAS benchmarks. Finally, tuning a NAS algorithm's hyperparameters can make it dramatically better, but transferring such hyperparameters across benchmarks often fails. This analysis strongly suggests adapting the empirical standard of the field, to stop focusing too much on smaller NAS benchmarks like NAS-Bench-101 and NAS-Bench-201, and rather also embrace larger and novel NAS benchmarks for non object classification tasks (Klyuchnikov et al., 2020; Mehrotra et al., 2021; Duan et al., 2021). While substantial differences across NAS search spaces has so far made it very hard to use many NAS benchmarks, we showed a way out of this dilemma by introducing an easy-to-use, unified benchmark suite that we hope will facilitate reproducible, generalizable, and rapid NAS research.

For future work, `NAS-Bench-Suite` can benefit from additional options, such as distributed training. Furthermore, although practitioners using NAS-Bench-Suite have the option to choose their own hand-picked subset of the 25 tasks based on their specific application, it would be useful to define representative subsets of the benchmarks in NAS-Bench-Suite based on application type.

## 8 ETHICS STATEMENT

Our work gives a large scale evaluation of generalizability in NAS and then proposes a new benchmark suite for NAS. The goal of our work is to make it faster and more accessible for researchers to run generalizable and reproducible NAS experiments. Specifically, the use of tabular and surrogate NAS benchmarks allow researchers to simulate NAS experiments cheaply on a CPU, rather than requiring a GPU cluster, reducing the carbon footprint of NAS research (Patterson et al., 2021; Hao, 2019). This is especially important since the development stage of NAS research may be extremely computationally intensive without the use of NAS benchmarks (Zoph & Le, 2017; Real et al., 2019). Our work is a tool for the NAS community, which facilitates NAS research that may be used for positive impacts on society (for example, algorithms that reduce $CO_2$ emissions (Rolnick et al., 2019)) or negative impacts on society (for example, models that discriminate or exclude groups of people). Due to the increase in conversations about ethics and societal impacts in the AI community (Hecht et al., 2018), we are hopeful that the applications of our work will have a net positive impact on society.

### ACKNOWLEDGMENTS AND DISCLOSURE OF FUNDING

FH and his group acknowledge support by the German Federal Ministry of Education and Research (BMBF, grant RenormalizedFlows 01IS19077C and grant DeToL), the Robert Bosch GmbH, the European Research Council (ERC) under the European Union Horizon 2020 research and innovation programme through grant no. 716721, and by TAILOR, a project funded by EU Horizon 2020 research and innovation programme under GA No 952215. This research was funded by the Deutsche Forschungsgemeinschaft (DFG, German Research Foundation) under grant number 417962828. We thank Danny Stoll and Falak Vora for their helpful contributions to this project.

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

Table 3: List of abbreviations used in the text.

| Abbreviation | Full |
| --- | --- |
| RS | Random Search |
| RE | Regularized Evolution |
| LS | Local Search |
| NPENAS | Neural Predictor Guided Evolution for Neural Architecture Search |
| BANANAS | Bayesian Optimization with Neural Architectures for Neural Architecture Search |
| GP | Gaussian Process |
| NAO | Neural Architecture Optimization |
| RF | Random Forest |
| XGB | XGBoost (Extreme Gradient Boosting) |
| BOHAMIANN | Bayesian Optimization with Hamiltonian Monte Carlo Artificial Neural Networks |
| NB-101 | NAS-Bench-101 |
| NB-201 | NAS-Bench-201 |
| NB-301 | NAS-Bench-301 |
| NB-ASR | NAS-Bench-ASR (Automated Speech Recognition) |
| NB-NLP | NAS-Bench-NLP (Natural Language Processing) |
| NB-MR | NAS-Bench-MR (Multi-Resolution) |
| TNB-Micro | TransNAS-Bench, Micro search space |
| TNB-Macro | TransNAS-Bench, Macro search space |
| DARTS | Differentiable Architecture Search |
| GDAS | Gradient-based search using Differentiable Architecture Sampler |
| DrNAS | Dirichlet Neural Architecture Search |

## A  BEST PRACTICES FOR NAS RESEARCH

There have been a few recent works which have called for improving the reproducibility and fairness in experimental comparisons in NAS research (Li & Talwalkar, 2019; Ying et al., 2019; Yang et al., 2020). This led to the release of a NAS best practices checklist (Lindauer & Hutter, 2020). We address each part of the checklist.

**Best practices for releasing code.**  For each NAS benchmark that we used, the code for the training pipeline and the search space is already publicly available. Since we used NAS benchmarks for all of our experiments, we did not evaluate the architectures ourselves. All of the code for the NAS methods, including the hyperparameters, are available at `https://github.com/automl/naslib`. We discuss the choices of hyperparameters in Appendix C.

**Best practices for comparing NAS methods.**  Since we made use of NAS benchmarks, all of the details for training the architectures are fixed across NAS methods. We included baselines such as random search and local search in our experiments in Section 4. We averaged 20 trials and 100 or more hyperparameter configurations for each experiment, and the choice of seeds (0-19) and hyperparameter configurations is available at `https://github.com/automl/naslib`.

**Best practices for reporting important details.**  We reported how we tuned the hyperparameters of NAS methods in Section 4. We included all details of our experimental setup in Section 4 and C.

## B  DETAILS FROM SECTION 2

This section is an extension of the "NAS benchmarks" part of Section 2, with additional details for each NAS benchmark, as well as a more comprehensive list of NAS benchmarks. For an extension of Table 1, which contains a more comprehensive list, see Table 4.

**NAS benchmarks.**   The first tabular NAS benchmark to be released was NAS-Bench-101 (Ying et al., 2019). This benchmark consists of a cell-based search space of $423\,624$ architectures with a fixed macro structure. NAS-Bench-101 comes with precomputed validation and test accuracies at epochs 4, 12, 36, and 108 from training on CIFAR-10. The cell-based search space of NAS-Bench-101 consists of five nodes which can take on any DAG structure with at most seven edges. Each node can take on one of three operations. Since training with stochastic gradient descent is random, all architectures were trained three times with different seeds and therefore have three sets of accuracies.

Since NAS-Bench-101 architectures contain a variable amount of nodes, it is not possible to evaluate one-shot algorithms. Therefore, NAS-Bench-1Shot1 (Zela et al., 2020b) defines three subsets of NAS-Bench-101 which allow one-shot algorithms to be run. The largest subset size in NAS-Bench-1Shot1 is $363\,648$.

NAS-Bench-201 is the second tabular NAS benchmark. It consists of a cell which is a complete directed acyclic graph over 4 nodes. Therefore, there are $\binom{4}{2} = 6$ edges. Each edge can take on one of five operations (note that this is in contrast to NAS-Bench-101, in which the *nodes* are operations). The search space consists of $5^6 = 15,625$ neural architectures, although due to none and identity operations, the number of non-isomorphic architectures is $6\,466$. Each architecture has precomputed train, validation, and test losses and accuracies for 200 epochs on CIFAR-10, CIFAR-100, and ImageNet-16-120. As in NASBench-101, on each dataset, each architecture was trained three times using different random seeds.

NATS-Bench (Dong et al., 2021) is an extension of NAS-Bench-201 which also varies the macro architecture. Specifically, a search space of $32\,768$ architectures with varying size were trained across three datasets for three seeds.

The first non computer vision NAS benchmark to be released was NAS-Bench-NLP (Klyuchnikov et al., 2020). Its search space consists of a DAG of up to 24 nodes, each of which can take on one of seven operations and can have at most three incoming edges. With a size of at least $10^{53}$, NAS-Bench-NLP is currently the largest NAS benchmark. $14\,322$ of the architectures were trained on Penn Tree Bank (Mikolov et al., 2010) for 50 epochs. Since only a fraction of architectures were trained, NAS-Bench-NLP is not queryable.

The DARTS (Liu et al., 2019b) search space with CIFAR-10 is arguably the most popular NAS benchmark. The search space contains $10^{18}$ architectures, consisting of two cells, each of which has six nodes. Each node has exactly two incoming edges, and each edge can take one of eight operations. Recently, $60\,000$ of the architectures were trained for 100 epochs and used to create NASBench-301 Siems et al. (2020), the first surrogate NAS benchmark. The authors released pretrained surrogates created using XGBoost Chen & Guestrin (2016) and graph isomorphism networks (Xu et al., 2019).

NAS-Bench-ASR (Mehrotra et al., 2021) is a tabular NAS benchmark for automatic speech recognition. The search space consists of $8\,242$ architectures trained on the TIMIT dataset. The search space consists of four nodes, with three main edges that can take on one of six operations, and six skip connection edges, which can be set to on or off.

NAS-Bench-111, NAS-Bench-311, and NAS-Bench-NLP11 (Yan et al., 2021) were recently released as surrogate benchmarks that extend NAS-Bench-101, NAS-Bench-301, and NAS-Bench-NLP by predicting the full learning curve information. In particular, none of NAS-Bench-101, NAS-Bench-301, and NAS-Bench-NLP allow the validation accuracies to be queried at arbitrary epochs, which is necessary for multi-fidelity NAS techniques such as learning curve extrapolation (Baker et al., 2018; Klein et al., 2017). The surrogates used to create NAS-Bench-111, NAS-Bench-311, and NAS-Bench-NLP11 include singular value decomposition and noise modeling (Yan et al., 2021).

TransNAS-Bench (Duan et al., 2021) is a tabular NAS benchmark consisting of two separate search spaces (cell-level and macro-level) and seven tasks including pixel-level prediction, regression, and

Table 4: A comprehensive overview of NAS benchmarks.

| Benchmark | Size | Queryable Tab. | Surr. | LCs | Macro | Type | #Tasks | NAS-Bench-Suite |
|---|---|---|---|---|---|---|---|---|
| NAS-Bench-101 | 423k | ✓ | | | | Image class. | 1 | ✓ |
| NAS-Bench-201 | 6k | ✓ | | ✓ | | Image class. | 3 | ✓ |
| NATS-Bench | 6k | ✓ | | ✓ | ✓ | Image class. | 3 | ✓ |
| NAS-Bench-NLP | $10^{53}$ | | | ✓ | | NLP | 1 | ✓ |
| NAS-Bench-1Shot1 | 364k | ✓ | | | | Image class. | 1 | ✓ |
| NAS-Bench-301 | $10^{18}$ | | ✓ | | | Image class. | 1 | ✓ |
| NAS-Bench-ASR | 8k | ✓ | | | ✓ | ASR | 1 | ✓ |
| TransNAS-Bench | 7k | ✓ | | ✓ | ✓ | Var. CV | 14 | ✓ |
| NAS-Bench-111 | 423k | | ✓ | ✓ | | Image class. | 1 | ✓ |
| NAS-Bench-311 | $10^{18}$ | | ✓ | ✓ | | Image class. | 1 | ✓ |
| NAS-Bench-NLP11 | $10^{53}$ | | ✓ | ✓ | | NLP | 1 | ✓ |
| NAS-Bench-MR | $10^{23}$ | | ✓ | | ✓ | Var. CV | 9 | ✓ |
| NAS-Bench-360 | Var. | | | | ✓ | Var. | 30 | |
| NAS-Bench-Macro | 6k | ✓ | | | ✓ | Image class. | 1 | |
| HW-NAS-Bench (201) | 6k | ✓ | | ✓ | | Image class. | 3 | |
| HW-NAS-Bench (FBNet) | $10^{21}$ | | | | | Image class. | 1 | |

self-supervised tasks. The-cell level search space of TransNAS-Bench is similar to that of NAS-Bench-201, but with 4 choices of operations per edge, hence $4\,096$ architectures altogether. The macro-level search space instead has a flexible macro skeleton with variable number of blocks, locations to down-sample feature maps, and locations to raise the channels, leading to a total of $3\,256$ architectures.

NAS-Bench-MR (Ding et al., 2021) is a surrogate NAS benchmark which evaluates nine settings total, across four datasets: ImageNet50-1000, Cityscapes, KITTI, and HMDB51. NAS-Bench-MR consists of a single search space of size $10^{23}$, and for each of the nine settings, $2\,500$ architectures were trained, to create nine different surrogates for each of the nine settings.

NAS-Bench-360 (Tu et al., 2021) is a very recent benchmark suite which gives NAS benchmarks for ten different datasets, including tasks that are novel for NAS such as spherical projection, fluid dynamics, DNA sequencing, medical imaging, surface electromyography, and cosmic ray detection. The tasks are carried out on three different search spaces based on Wide ResNet (He et al., 2016), DARTS (Liu et al., 2019b), and DenseNAS (Fang et al., 2020). However, a drawback of NAS-Bench-360 is that none of the NAS benchmarks are queryable. Therefore, evaluating NAS algorithms on these benchmarks requires 1 to $100+$ GPU-hours of runtime (Tu et al., 2021).

NAS-Bench-Macro (Su et al., 2021) is a NAS benchmark which focuses on the macro search space. It consists of 6561 pretrained architectures on CIFAR-10. The search space consists of 8 layers, each with 3 choices of blocks.

HW-NAS-Bench is a NAS benchmark focusing on hardware-aware neural architecture search. It gives the measured/estimated hardware-cost for all architectures in NAS-Bench-201 and FBNet (Wu et al., 2019) on six hardware devices, including commercial edge, FPGA, and ASIC devices. HW-NAS-Bench can be used alongside NAS-Bench-201 for the full information on hardware cost and model accuracy for all architectures in NAS-Bench-201.

## C  DETAILS FROM SECTION 4

### C.1  NAS ALGORITHM IMPLEMENTATION DETAILS

Here, we give implementation details for all algorithms that we compared in Section 4. We made an effort to keep the implementations as close as possible to the original implementation. We start with the black-box optimizers. For a list of the default hyperparameters and hyperparameter ranges, see `https://github.com/automl/NASLib`.

- **Random search.** Random search is the simplest baseline for NAS Li & Talwalkar (2019); Sciuto et al. (2020). It draws architectures at random and then returns the best architecture.

- **Local search.** Another baseline, local search has been shown to perform well on multiple NAS benchmarks (White et al., 2021b; Ottelander et al., 2021; Siems et al., 2020). It works by evaluating all architectures in the neighborhood of the current best architecture found so far. The neighborhood of an architecture is the set of architectures which differ by one operation or edge. We used the implementation from White et al. (White et al., 2021b).

- **Regularized evolution.** This algorithm (Real et al., 2019) consists of iteratively mutating the best architectures drawn from a sample of the most recent architectures evaluated. A mutation is defined by randomly changing one operation or edge. We used the NAS-Bench-101 (Ying et al., 2019) implementation.

- **BANANAS.** This NAS algorithm (White et al., 2021a) uses Bayesian optimization with an ensemble of three predictors as the surrogate. We use the code from the original repository, but using PyTorch for the MLPs instead of Tensorflow. We use the adjacency matrix encoding instead of the path encoding, since the path encoding does not scale to large search spaces such as NAS-Bench-NLP. We use variational sparse GPs (Titsias, 2009) in the ensemble of predictors, since this was shown in prior work to perform well and have low runtime (White et al., 2021c).

- **NPENAS.** This algorithm (Wei et al., 2020) is based on predictor-guided evolution. It iteratively chooses the next architectures by mutating the most recent architectures in a random sample of the population to create a set of candidate architectures, and then using a predictor to pick the architectures with the highest expected accuracy. Again, we use variational sparse GP (Titsias, 2009) as the predictor.

Now we describe the performance predictors. For each method, we used the adjacency one-hot encoding (White et al., 2020).

- **BOHAMIANN.** BOHAMIANN (Springenberg et al., 2016) is a Bayesian inference prediction method which uses stochastic gradient Hamiltonian Monte Carlo (SGHMC) in order to sample from a Bayesian Neural Network. We use the original implementation from the `pybnn` package.

- **GP.** Gaussian Process (GP) (Rasmussen, 2003) is a popular surrogate often used with Bayesian optimization (Frazier, 2018; Snoek et al., 2012). Every feature has a joint Gaussian distribution. We use the `Pyro` implementation (Bingham et al., 2019).

- **NAO.** Neural Architecture Optimization is a NAS method that uses an encoder-decoder (Luo et al., 2018). The encoder is a feedfoward neural network, and the decoder is an LSTM and attention mechanism. We used the implementation from SemiNAS (Luo et al., 2020).

- **Random Forest.** Random forests (Breiman, 2001) are ensembles of decision trees. Random forests have been used as model-based predictors in NAS (Siems et al., 2020; White et al., 2021c). We use the Scikit-learn implementation (Pedregosa et al., 2011).

- **XGBoost.** eXtreme Gradient Boosting (XGBoost) (Chen & Guestrin, 2016) is a popular gradient-boosted decision tree which has been used in NAS (Siems et al., 2020; White et al., 2021c). We used the original code (Chen & Guestrin, 2016).

For one-shot methods, we mainly focus on the differentiable architecture search approaches, and select popular algorithms like DARTS (Liu et al., 2019b), GDAS (Dong & Yang, 2019) and DrNAS (Chen et al., 2021) in our paper.

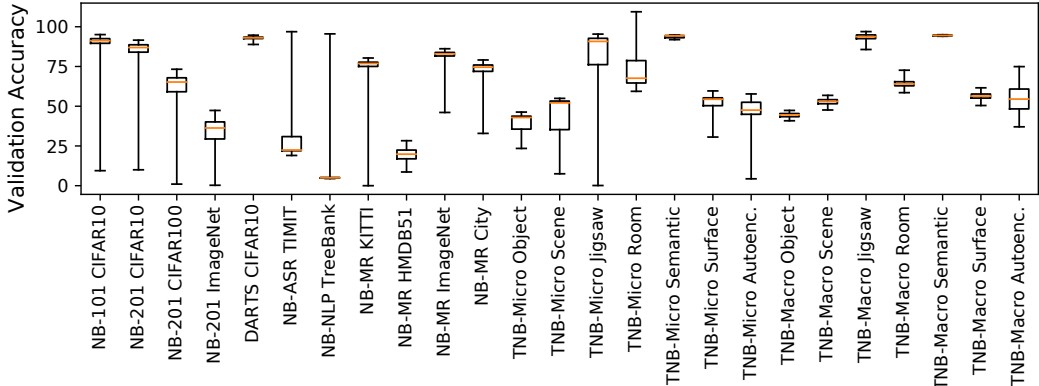

Figure 7: Validation accuracy box plots for each NAS benchmark. The whiskers represent the minimum and maximum accuracies in each search space. For NAS-Bench-NLP, perplexity is used instead of validation accuracy, and three datasets of TransNAS-Bench do not use accuracy: Surface Normal uses SSIM, Autoencoding uses SSIM, and Room Layout uses negative loss. These are in accordance with the metrics used in the original work. Finally, in the case of extremely large search spaces such as DARTS and NAS-Bench-NLP, the statistics are computed only with respect to the tens-of-thousands of precomputed architectures.

- **DARTS.** DARTS (Liu et al., 2019b) is the first work on differentiable architecture search. Compared to normal one-shot method that has a one-hot encoding to select one architecture out of different choices, it uses a vector, a.k.a. architecture parameters, where each element ranges from 0 to 1 as probability. During training, it sums all branches as a weight summation. After the search, the final architecture is converted by selecting the branch that with highest weight.

- **GDAS.** Since differentiable architecture search suffers from unstable training compared to traditional one-shot methods, GDAS (Dong & Yang, 2019) bridges the gap. Instead of having a weighted summation of all paths, it discretizes the paths during training to select the path with highest probability, while the rest of algorithms remains similar as original DARTS.

- **DrNAS.** Dirichlet architecture search (DrNAS) (Chen et al., 2021) is another attempt to solve the instability issue of differentiable architecture search. This work treats the continuously relaxed architecture weights as a random variable, which is modeled by a Dirichlet distribution. To this end, the Direchlet parameters can be updated by the traditional differentiable architecture search optimizer easily in the end-to-end manner.

# D ADDITIONAL EXPERIMENTS

In this section, we give additional statistics, algorithm, and insight experiments to augment Sections 3 and 4.

## D.1 ADDITIONAL STATISTICS EXPERIMENTS

We start by giving additional experiments on the statistics of NAS benchmarks, to supplement Section 3. In Figure 7, we give the full box plots, extending Figure 2. We also add new statistics: in Figure 8, we plot the average time to train an architecture for each NAS benchmark. In Figure 9, we plot the average neighborhood size for each NAS benchmark. Note that for some NAS benchmarks, the neighborhood size is fixed, and for other NAS benchmarks, the neighborhood size varies.

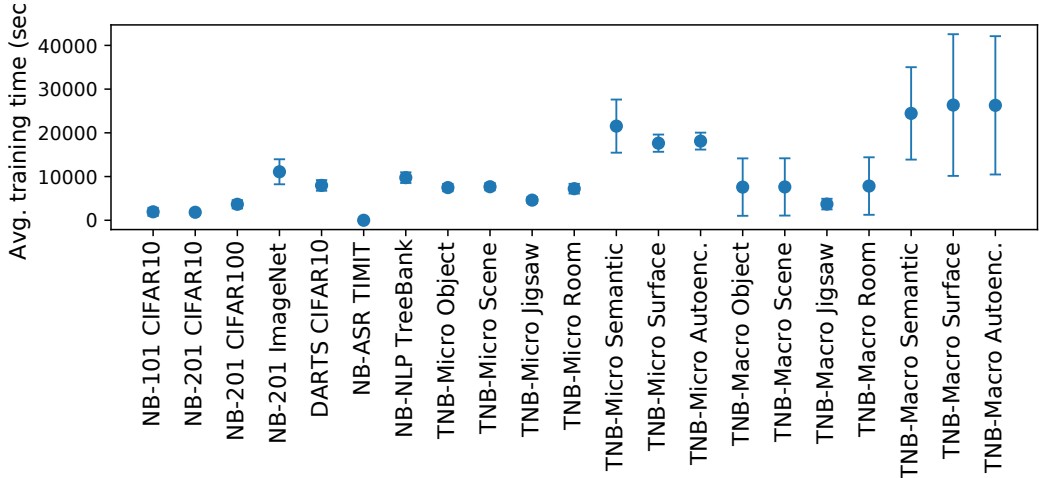

Figure 8: Average time to train an architecture for each NAS benchmark.

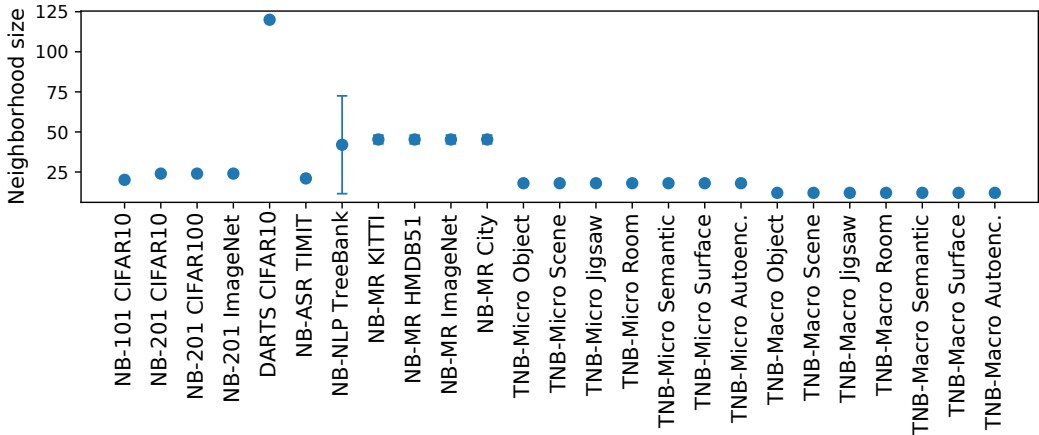

Figure 9: Average neighborhood size for each NAS benchmark. Note that for some NAS benchmarks, the neighborhood size is fixed, and for other NAS benchmarks, the neighborhood size varies.

## D.2 ADDITIONAL ALGORITHM EXPERIMENTS

Next, we give additional experiments from Section 4. In Figure 10, we give the full performance predictor and black-box results, extending Figure 4.

Recall that in Section 4, we assessed the transferability of hyperparameters by tuning algorithms on NAS benchmark $A$, and evaluating the performance of the tuned method on NAS benchmark $B$, compared to the performance of the best hyperparameters from NAS benchmark $B$. The results were plotted in Figure 5. In Tables 5 and 6, to present the results in another format, we give the raw values from these experiments. All values are averaged over each search space. For example, the three rows for NAS-Bench-201 were averaged into one row, and similarly for the columns.

Furthermore, in Tables 7 and 8, we give summary statistics from this experiment. For each search space, we compute the transferability of hyperparameters on average to or from all other search spaces. For performance predictors, we find that hyperparameters from NAS-Bench-MR transfer the least well to or from other search spaces. NAS-Bench-NLP transfers the best. For black-box algorithms, NAS-Bench-MR transfers the worst, and DARTS transfers the best. Therefore, it is safest for practitioners to tune their techniques on NAS-Bench-NLP and DARTS before deploying them in a new setting. It is not safe to tune techniques on other benchmarks such as NAS-Bench-MR.

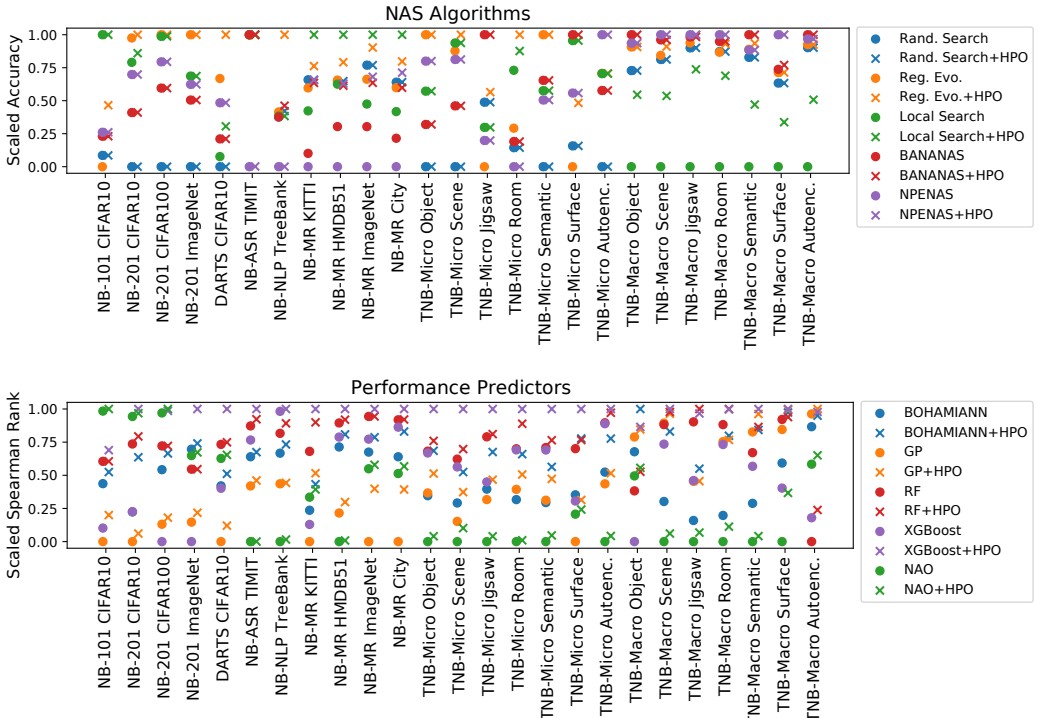

Figure 10: Relative performance of black-box algorithms (top) and performance predictors (bottom) across NAS benchmarks. The solid circle shows the performance of the algorithm with default hyperparameters, while the cross shows performance after hyperparameter optimization.

While all of the hyperparameter transfer experiments to this point have focused on the optimal hyperparameters, we now present two more experiments that focus on the hyperparameters on average. In Figure 11, for search spaces A and B, we compute the Kendall Tau rank correlation of the ranking lists for *all* hyperparameters on search spaces A and B.

Finally, in Table 9, we present "leave one out" experiments for all search spaces. For a search space A, the best hyperparameter setting on average over all search spaces except A is computed, and compared to the performance of the best hyperparameters for A. This is similar to Table 7 in that it provides a summary of the average transferability of the hyperparameters for each search space. However, the leave one out experiments are focused more on the performance of hyperparameters on average, while Table 7 is focused on transferability of the optimal hyperparameters.

### D.3 A GUIDE TO INTERPRETING HYPERPARAMETER TRANSFER EXPERIMENTS

Throughout Sections 4 and D.2, we presented several different analyses and summaries on the extent to which hyperparameters transfer from one search space to another. In this section, we give a guide for interpreting the results.

First, practitioners interested in the transfer of hyperparameters which are *optimally trained* on one search space, should focus on Figure 5 and Tables 7 and 8, because these figures and tables express the regret of hyperparameters tuned on one search space and evaluated on others. On the other hand, practitioners interested on how hyperparameters overall transfer from one search space to others (not just optimal), should focus on Figure 11 and Table 9, because these represent the average transferability of all sets of hyperparameters that we tried.

Practitioners interested in the specific transfer from one search space (or setting) to another, should focus on our matrix results, Figures 5 and 11. Practitioners interested in the general transferability on average to or from one search space to others, should focus on the summary tables, Tables 7, 8, and 9.

Table 5: Raw values from the performance predictor hyperparameter transferability experiment from Figure 5 (left). Each search space has 0-1 scaling done to fairly compare trends between search spaces. Results are weighted by search space. E.g., each of the three NAS-Bench-201 benchmarks are averaged into one row/column.

|         | NB-101 | NB-201 | DARTS | NB-ASR | NB-NLP | NB-MR | TNB-101 |
|---------|--------|--------|-------|--------|--------|-------|---------|
| NB-101  | .00    | .42    | .28   | .46    | .48    | .21   | .41     |
| NB-201  | .43    | .02    | .29   | .11    | .11    | .32   | .09     |
| DARTS   | .08    | .47    | .00   | .48    | .34    | .26   | .41     |
| NB-ASR  | .28    | .23    | .31   | .00    | .07    | .32   | .12     |
| NB-NLP  | .25    | .31    | .27   | .09    | .00    | .36   | .15     |
| NB-MR   | .19    | .39    | .28   | .42    | .47    | .20   | .40     |
| TNB-101 | .37    | .13    | .41   | .16    | .15    | .40   | .15     |

Table 6: Raw values from the black-box algorithm hyperparameter transferability experiment from Figure 5 (right). Each search space has 0-1 scaling done to fairly compare trends between search spaces. Results are weighted by search space. E.g., each of the three NAS-Bench-201 benchmarks are averaged into one row/column.

|         | NB-101 | NB-201 | DARTS | NB-ASR | NB-NLP | NB-MR | TNB-101 |
|---------|--------|--------|-------|--------|--------|-------|---------|
| NB-101  | .00    | .25    | .25   | .00    | .25    | .75   | .27     |
| NB-201  | .25    | .22    | .33   | .75    | .50    | .83   | .28     |
| DARTS   | .25    | .33    | .00   | .75    | .50    | .50   | .23     |
| NB-ASR  | .75    | .50    | .50   | .00    | .75    | .25   | .50     |
| NB-NLP  | .07    | .28    | .28   | .73    | .00    | .64   | .28     |
| NB-MR   | .86    | .80    | .70   | .42    | .84    | .32   | .76     |
| TNB-101 | .27    | .28    | .23   | .73    | .48    | .70   | .28     |

## D.4 INSIGHT EXPERIMENTS

Now we present experiments that give new insights into NAS algorithms and search spaces.

First, we run experiments to test the following hypothesis: larger search spaces have smaller interquartile ranges (IQR), because a larger cell size gives most architectures a chance to have performant operations. For example, for a search space of size three, some architectures will have two or three convolution operations, and some architectures will have two or three pooling operations, creating a large IQR. But for a search space of size 10, the vast majority of architectures will have a good mix of convolution and pooling operations, creating a comparatively lower IQR. We run this experiment on NAS-Bench-101 and NAS-Bench-201. See Figure 12. We find that in all benchmarks, there is a strict negative correlation between number of operations and IQR.

Finally, we compute correlations between the relative ranking of each NAS technique and properties of the search spaces, such as total size and neighborhood size. See Figure 10. The largest correlations we find are as follows:

- GP performs comparatively much better on search spaces with small neighborhood sizes.

- When tuned, RF and XGBoost perform comparatively much better on large search spaces and also search spaces with large neighborhood sizes.

- Surprisingly, BOHAMIANN performs comparatively better for large neighborhood sizes when not tuned, and comparatively better for small neighborhood sizes when tuned.

- Default regularized evolution performs comparatively much better on search spaces with small neighborhood sizes.

Table 7: Summaries from the performance predictor transferability experiment from Figure 5 (left). Each value in "transfer to" is the average of the corresponding row from Figure 5. Each value in "transfer from" is the average of the corresponding column. Therefore, for each search space, we have a measure of the extent to which hyperparameters can transfer to or from other search spaces, on average.

|  | NB-101 | NB-201 | DARTS | NB-ASR | NB-NLP | NB-MR | TNB-101 |
|---|---|---|---|---|---|---|---|
| Transfer to | 0.376 | 0.229 | 0.340 | 0.224 | 0.239 | 0.393 | 0.293 |
| Transfer from | 0.268 | 0.328 | 0.307 | 0.288 | 0.270 | 0.345 | 0.287 |

Table 8: Summaries from the black-box algorithm transferability experiment from Figure 5 (right). Each value in "transfer to" is the average of the corresponding row from Figure 5. Each value in "transfer from" is the average of the corresponding column. Therefore, for each search space, we have a measure of the extent to which hyperparameters can transfer to or from other search spaces, on average.

|  | NB-101 | NB-201 | DARTS | NB-ASR | NB-NLP | NB-MR | TNB-101 |
|---|---|---|---|---|---|---|---|
| Transfer to | 0.461 | 0.528 | 0.428 | 0.542 | 0.381 | 0.784 | 0.495 |
| Transfer from | 0.407 | 0.444 | 0.383 | 0.731 | 0.555 | 0.666 | 0.433 |

- Local search performs comparatively better on small search spaces and also search spaces with small neighborhood sizes.

# E    DETAILS FROM SECTION 5

In this section, we give more details on the API for `NAS-Bench-Suite`.

When designing `NAS-Bench-Suite`, we strove for both minimalism and generalism. In order to add a new benchmark in `NAS-Bench-Suite`, one has to first define the computational graph using the NetworkX API. This graph encompasses high-level abstractions such as the `add_node` and `add_edges_from` methods. If we are implementing a tabular or surrogate benchmark as the ones used in Section 4, a `get_dataset_api` function needs to be implemented, which is used as an interface to the original pre-computed benchmark data. The PyTorch computational graph is generated via the `adapt_search_space` method of the NAS algorithms or performance predictors. For instance, it can determine if operation choices in edges should be combined as a mixed operation (as done in DARTS (Liu et al., 2019b)) or if they should be categorical choices from which the NAS algorithms sample. Afterwards, the graph instance is stored as an attribute of the optimizer instance. The `Trainer`, which runs the optimization loop, interacts only with the optimizer (see line 16 in Snippet 1).

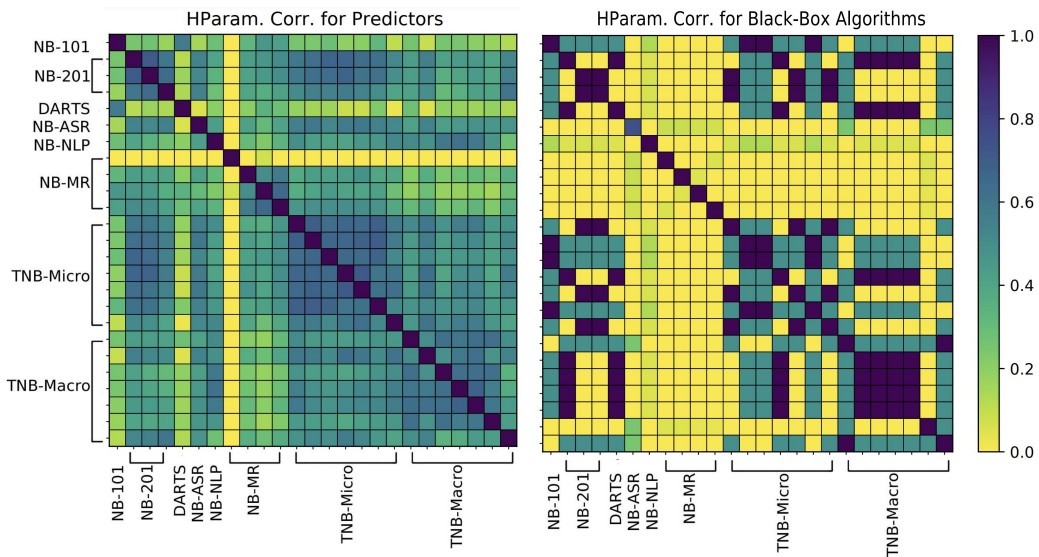

Figure 11: Transferability results for predictors (left) and black-box algorithms (right). Row $i$, column $j$ denotes the Kendall Tau rank correlation of the performance of hyperparameters between search spaces $i$ and $j$. For abbreviations, see Table 3.

Table 9: Leave one out experiments for performance predictors. For a search space A, the best hyperparameter setting on average over all search spaces except A is computed, and compared against the best hyperparameter setting of search space A when transferring to (or from) search space A.

|  | NB-101 | NB-201 | DARTS | NB-ASR | NB-NLP | NB-MR | TNB-101 |
|---|---|---|---|---|---|---|---|
| Transfer to | 0.372 | 0.37 | 0.48 | 0.195 | 0.185 | 0.445 | 0.252 |
| Transfer from | 0.226 | 0.16 | 0.223 | 0.058 | 0.029 | 0.241 | 0.056 |

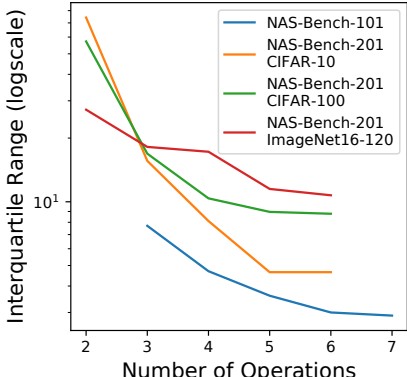

Figure 12: Interquartile ranges of subsets of NAS-Bench-101 and NAS-Bench-201 as a function of the number of operations. As the size of the search space increases, the interquartile range decreases.

Table 10: Correlation insights for five NAS algorithms (left) or five performance predictors (right). Kendall Tau rank correlations are computed between properties of the search spaces (search space size or neighborhood size) and the relative ranking list for predictors or NAS algorithms, with or without HPO. Since a ranking list is used, a *negative* correlation means a *positive* correlation between the search space property and algorithmic performance.

| | Performance Predictors | | | | | NAS Algorithms | | | | |
| | BOHAM. | GP | RF | XGB | NAO | RS | RE | BANANAS | LS | NPENAS |
|---|---|---|---|---|---|---|---|---|---|---|
| Default vs. SS size | -.07 | .25 | -.16 | -.39 | .20 | -.29 | -.21 | -.15 | .48 | .26 |
| HPO vs. SS size | .15 | .05 | -.51 | -.39 | .20 | -.26 | -.21 | -.05 | .21 | .35 |
| Default vs. Nbhd. size | -.33 | .45 | -.26 | .00 | .00 | -.10 | -.51 | .25 | .37 | .16 |
| HPO vs. Nbhd. size | .35 | .37 | -.39 | -.51 | .00 | -.16 | -.31 | .05 | .31 | .05 |

