# OpenReview forum: "NAS-Bench-Suite: NAS Evaluation is (Now) Surprisingly Easy"
_ICLR.cc/2022/Conference — ICLR 2022 Poster_

### Official Review · Reviewer_Gku7 · 2021-10-21

**Correctness:** 3
**Technical Novelty And Significance:** 1
**Empirical Novelty And Significance:** 4
**Recommendation:** 8
**Confidence:** 5

**Main Review:**

The results are interesting and very useful for the community as it is - and the authors should be commended for their effort.
However the work lacks deeper insights from the experiments performed that could guide the community towards better methods. For example: [Fig.2] Are there identifiable components of architectures that lead to the interquartile range differences between the CIFAR10 benchmarks (NB101, NB201, DARTS)? Or [Table 2] What makes XGBoost generalize well where others don't?

Sec.4.2: Transfer is always expected to lead to a performance drop. Is there evidence that NAS is especially bad in this respect, compared to HP transfer in other settings? (say, transferring ResNets HPs from one dataset to another)

Fig.5: Since the figure contains a lot of information, could the authors provide some useful summary statistics in addition?

Minor: clarify whether $i$ and $j$ are columns or rows in Fig.5.

**Summary Of The Paper:**

This paper proposes a unified interface for access to a collection of Neural Architecture Search (NAS) benchmarks. With experiments performed on across multiple search spaces and datasets, the authors show that some conclusions drawn from a small subset of benchmarks do not generalize across diverse datasets and tasks.

**Summary Of The Review:**

In my view the two main contributions of this work are (1) showcasing the limitations of drawing conclusions from single benchmarks; (2) providing the community with a larger analysis tool.

I expect that the community will welcome these resources and thereby **recommend the paper is accepted** (provisional score: 8)

---

> ### Author Response · Authors · 2021-11-16
> **Thank you for the great suggestions. We have incorporated them into the paper.**
>
> We thank the reviewer for their insightful review. We reply to the comments below.
>
> **”the work lacks deeper insights from the experiments performed that could guide the community towards better methods”**
>
> We agree, and so we 1) investigated your questions, and 2) showed that we can combine our statistics experiments and algorithm experiments to present deeper insights to the community.
>
> - We explored your question on whether there are “identifiable components of architectures that lead to the interquartile range differences between the CIFAR10 benchmarks.” We found that a large factor in the interquartile range (IQR) differences is simply the *cell size*, because search spaces with large cell sizes tend to have most architecture achieve similar accuracy. For example, for a search space of size three, some architectures will have two or three convolution operations, and some architectures will have two or three pooling operations, creating a large range of accuracies. But for a search space of size 10, the vast majority of architectures will have a good mix of convolution and pooling operations, creating a comparatively lower range of accuracies. We tested this hypothesis by computing the IQR of subsets of NAS-Bench-101 and NAS-Bench-201 (from 1 to 6 operations), and we found that in all cases, there is a strict negative correlation between number of operations and IQR. See Fig. 12.
>
> - Your comment about the generalizability of XGBoost is a great point. While a thorough answer to this is a good question for future work, we found that our results are already in line with prior work that show XGBoost (and generally gradient-boosted trees) generalize well compared to other algorithms including neural networks [1, 2, 3].
>
> - Finally, in order to give deeper insights that can help the community, we combined our statistics experiments from Section 3 with our algorithm experiments from Section 4, to compute trends between algorithmic performance and properties of the search space. We present our findings in Table 10. The strongest correlations are as follows:
>   - Regularized evolution performs comparatively much better on search spaces with small neighborhood sizes.
>   - Local search performs comparatively better on small search spaces and also search spaces with small neighborhood sizes.
>   - When tuned, RF and XGBoost perform comparatively much better on large search spaces and also search spaces with large neighborhood sizes.
>
> **”Transfer is always expected to lead to a performance drop. Is there evidence that NAS is especially bad in this respect, compared to HP transfer in other settings? (say, transferring ResNets HPs from one dataset to another)”**
>
> This is a great point, and we ran additional experiments to help resolve this question. Fully answering the question would require additional experiments with ResNets, which is a great question for future work. However, in our update to the paper, we gave a much more thorough answer to *exactly what practitioners can expect when transferring NAS hyperparameters*. Specifically, we made the following additions (see also our [reply to reviewer xtz2](https://openreview.net/forum?id=0DLwqQLmqV&noteId=jJZ5RvTtWL-)):
> - To present the results in another format, we added tables of the raw values from Fig. 5 in the appendix (Tables 5 and 6). We also included summary statistics (Tables 7 and 8) and a guide for how practitioners should interpret all of these hyperparameter transfer results (Appendix Section C.3).
> - We added “leave one out” experiments for each search space, to show specifically how the best hyperparameters for each search space compares to the best average over all other search spaces (Table 9).
> - We ran a new experiment similar to Fig. 5: for search spaces x and y, we plot the rank correlation of the ranking lists for all hyperparameters on search spaces x and y (Fig. 11). This gives a measurement of transfer for the hyperparameters on average, compared to Fig. 5 which focused on measuring the transfer for the optimal hyperparameters.
>
> **”Fig.5: Since the figure contains a lot of information, could the authors provide some useful summary statistics in addition?”**
>
> Thanks for the suggestion. We added summary statistics of Fig. 5 to the updated paper (Tables 7 and 8). Overall, NAS-Bench-MR has the worst hyperparameter transfer to/from other search spaces, and NAS-Bench-NLP and DARTS have the best hyperparameter transfer.
>
> We thank the reviewer again for bringing up these points which we believe have further improved our paper. If you have any additional comments or questions, please let us know.
>
> [1] ​​Shavitt and Segal. Regularization learning networks: deep learning for tabular datasets. NeurIPS 2018.
>
> [2] Kadra et al. Well-tuned Simple Nets Excel on Tabular Datasets. NeurIPS 2021.
>
> [3] Bentéjac et al. A comparative analysis of gradient boosting algorithms. Artificial Intelligence Review, 2021.

---

> > ### Comment · Reviewer_Gku7 · 2021-11-18
> > **-**
> >
> > I thank the authors for providing the extra results. I think these enrich the paper and will be very useful for practitioners.

---

### Official Review · Reviewer_Ensh · 2021-11-02

**Correctness:** 4
**Technical Novelty And Significance:** 3
**Empirical Novelty And Significance:** 3
**Recommendation:** 8
**Confidence:** 4

**Main Review:**

There exist many NAS benchmarks currently across diverse tasks. Many NAS works conduct experiments on several benchmarks to demonstrate the effectiveness and believe.
However, researcher find that NAS algorithms may perform differently across these benchmarks. Sometimes even reverse conclusions are drawn.

This paper investigates this problem and points out that the differences may come from the different search space design, training pipeline and hyperparameters, via comprehensive experiments across many existing benchmarks. It finds that conclusions from a few benchmarks do not generalize well to other benchmarks

Further, the paper proposes NAS-Bench-Suite, a collection of NAS benchmarks, with a unified interface, which is easy-to-use, to facilitate reproducible, generalizable and rapid NAS research.

The motivation of the work is clear. The experimental analysis is convincing. The paper is well organized. The claim is sufficiently addressed by the method.

Overall, this is a good work for the NAS community.

**Summary Of The Paper:**

This paper investigates the drawbacks of current existing NAS benchmarks and the works that evaluate on them. Then it presents NAS-Bench-Suite, an extensible collection of NAS bench-marks which is easy-to-use to facilitate reproducible, generalizable and rapid NAS research.

**Summary Of The Review:**

I think the paper is of good quality and will contribute to the NAS community.

---

> ### Author Response · Authors · 2021-11-16
> **Thank you for the positive feedback.**
>
> Thank you for your thoughtful review. We are glad to hear that you found the paper of good quality and that you think it will contribute to the NAS community.
>
> We would also like to mention based on the reviews, we have updated the paper as follows (also see the [general comment](https://openreview.net/forum?id=0DLwqQLmqV&noteId=LijX9WCT5R) for further details).
> - We added new experiments regarding HPO transferability.
> - We added analyses that give experimental insights, such as the correlation between search space sizes and interquartile ranges, and the correlation of search space properties and the performance of NAS techniques.
> - We made improvements to the codebase, including new tutorials.
>
> If you have any additional questions or comments, please let us know.

---

### Official Review · Reviewer_2xvD · 2021-11-03

**Correctness:** 4
**Technical Novelty And Significance:** 2
**Empirical Novelty And Significance:** 3
**Recommendation:** 6
**Confidence:** 4

**Main Review:**

strengths:
- The paper is well motivated and written.
- This new benchmark can be used to do a more deep analysis for the NAS methods.
- Some interesting observations.

weaknesses:
- The technical part is somewhat weak.
- The table and figure captions are too short to explain themselves. Please at least explain the abbreviations used in the table or figure.

**Summary Of The Paper:**

The authors collected most of the existing NAS benchmarks to construct a new benchmark.
A unified API is provided to use these existing search spaces and architecture datasets. Based on this the authors re-analyze some NAS algorithms on this new large and comprehensive benchmark and have some interesting observations.

**Summary Of The Review:**

The authors did much engineering efforts and emprical analysis, whereas the technical novelty is a little bit weak.

---

> ### Author Response · Authors · 2021-11-16
> **Thank you for your review.**
>
> Thank you for reviewing our manuscript, and we are happy to hear that you found the paper to be well-motivated and well-written. We reply to your comments below.
>
> **”The table and figure captions are too short to explain themselves. Please at least explain the abbreviations used in the table or figure.”**
>
> Thank you for the suggestion. We expanded the captions for Table 2 and Figures 2, 3, 4, 5, and 6 to make them more easily understandable and more informative, and we created and referred to a table of abbreviations (Table 3) in the captions.
>
> **”The technical novelty is a little bit weak.”**
>
> We agree with the reviewer that our paper does not introduce new methods and may thus be perceived as having lower technical novelty than other types of papers. We respectfully point out that making available 25 different tabular NAS benchmarks through a unified interface, and running experiments that refute common implicit assumptions held by the community, are novel contributions that might be very useful to the community. We would also like to mention that experimental-analysis papers have been accepted to recent ICLR, ICML, and NeurIPS conferences, and some of them have public reviews and meta-reviews available here:
>
> [1] [Yang et al., ICLR 2020. NAS evaluation is frustratingly hard.](https://openreview.net/forum?id=HygrdpVKvr&noteId=C5rPJ2sLzP)
>
> [2] [Schmidt et al., ICML 2021. Descending through a Crowded Valley - Benchmarking Deep Learning Optimizers.](https://icml.cc/virtual/2021/poster/8645)
>
> [3] [White et al., NeurIPS 2021. How Powerful are Performance Predictors in Neural Architecture Search?](https://openreview.net/forum?id=6RB77-6-_oI&noteId=l5H13jujQKo)
>
> [4] [Ning et al., NeurIPS 2021. Evaluating Efficient Performance Estimators of Neural Architectures.](https://openreview.net/forum?id=Esd7tGH3Spl&noteId=dl91y63zePD)
>
> Thank you for your time, and please let us know if you have any further comments or questions.

---

### Official Review · Reviewer_xtz2 · 2021-11-04

**Correctness:** 4
**Technical Novelty And Significance:** 3
**Empirical Novelty And Significance:** 3
**Recommendation:** 8
**Confidence:** 3

**Details Of Ethics Concerns:**

I do not see any ethical concerns.

**Main Review:**

Strengths:
- Very important topic. Benchmarks are critical to assess performance of NAS algorithms.
- Provide statistics across a wide array of NAS benchmarks, such as the distribution of accuracy across the search space.
- They show that no NAS method clearly outperform all other method across all benchmarks. Some do better with HPO, but this is typically to expensive to computer for each benchmark.
- Tests 2 relevant questions. 1) Whether NAS results generalizes from the small 101 and 202 benchmarks to larger ones and 2) NAS have robust hyperparameters. The third question seems to be to closely related to the second one: 3) We can cheaply optimize hyperparameters of NAS on tabular benchmarks and reuse them on another benchmark..
    - They show that 1) NAS algorithms do not generalize well from small 101 and 202 benchmarks to larger ones,
    - 2) Hyperparameters are not robust for NAS algorithms.
    - 3) Is not true obviously since hyperparameters of NAS algorithms are not robust.
- They contribute a valuable wrapper for 25 benchmarks, although it is not clear to me how the search space representation with NetworkX allows for a dynamic search space definition.

Weaknesses:
- Out of the 3 questions, it seems to me the third one is to closely related to the first one. And the experimental answer to this question is not clearly outlined in the paper neither.
- I am quite concerned about the maintainability of the library. The benchmarks are bash scripts that will be difficult to maintain. Some even contain specific slurm configurations for the cluster of the researchers.

Random walk plot is interesting. I would be great to have a confidence interval or at least a standard deviation for each curve.

**Summary Of The Paper:**

This paper present a wrapper for 25 NAS benchmarks and provide insightful analysis on NAS and predictor performances across the benchmarks. They show, among other things, that no NAS method is best across all tasks, that hyperparameters of NAS are not robust and that small tabular benchmarks are not representative of NAS method performances on larger benchmarks.

**Summary Of The Review:**

The wrapper of benchmark is a valuable contribution by itself, but this paper illustrates well the benefit of such benchmarks by providing insightful analysis of NAS algorithms.

---

> ### Author Response · Authors · 2021-11-16
> **Thank you, we added these suggestions to the paper and code.**
>
> We thank the reviewer for their helpful review. We reply to the comments below.
>
> **”Out of the 3 questions, it seems to me the third one is too closely related to the first one. And the experimental answer to this question is not clearly outlined in the paper neither.”**
>
> Thank you for pointing this out. We have now updated the paper to make the three assumptions clear. A1 is focused on the *algorithms* transferring among NAS benchmarks, while A3 is focused on the *hyperparameters themselves* transferring among NAS benchmarks. For A1, we computed the average rank of all of the algorithms across all search spaces (Table 2), and for A3, we computed the performance of the best hyperparameters from search space x, when applied to search space y, for all x, y (Fig. 5). We agree that we could give even more experimental evidence for A3. Therefore, we added the following new experiments and updates:
> - To present the results in another format, we added tables of the raw values from Fig. 5 in the appendix (Tables 5 and 6). We also included summary statistics (Tables 7 and 8) and a guide for how practitioners should interpret all of these hyperparameter transfer results (Appendix Section C.3).
> - We added “leave one out” experiments for each search space, to show specifically how the best hyperparameters for each search space compares to the best average over all other search spaces (Table 9).
> - We ran a new experiment similar to Fig. 5: for search spaces x and y, we plot the rank correlation of the ranking lists for all hyperparameters on search spaces x and y (Fig. 11). This gives a measurement of transfer for the hyperparameters on average, compared to Fig. 5 which focused on measuring the transfer for the optimal hyperparameters.
>
> **”The benchmarks are bash scripts that will be difficult to maintain. Some even contain specific slurm configurations for the cluster of the researchers”**
>
> We agree and we have updated our code, although first we would like to clarify that nearly all of our code was written in Python, including all the runner files, and the bash scripts were used only to launch batch experiments in one command. We now made the following changes to the structure and documentation of our code.
> - We moved the bash scripts outside of the main directory, to a new folder called `scripts`, and we kept the individual `runner.py` scripts inside the main directory. This is a best practice (especially when we release the code publicly and make it pip installable).
> - We added tutorial notebooks for 1) getting started with the codebase and 2) understanding search spaces. These can be found under the `examples` subdirectory and are linked in the readme.
>
> **”Random walk plot is interesting. It would be great to have a confidence interval or at least a standard deviation for each curve.”**
>
> Thank you for the suggestion. We added confidence intervals to the random walk autocorrelation plots (Figure 3).
>
> Please let us know if you have any further questions or concerns.

---

### Author Response · Authors · 2021-11-16
**Updates following the reviewers’ comments.**

Dear reviewers and AC, we have addressed all of the suggestions made by the reviewers. We thank all of the reviewers once again, as these suggestions have further improved the quality of our paper. We give a list of the main additions below.
- We added new analyses for HPO transferability as follows:
  - Summary statistics for Fig. 5 (Tables 7 and 8).
  - “Leave one out” experiments for each search space (Table 9).
  - In Fig. 11, we show an experiment similar to Fig. 5, that computes the correlation of hyperparameter effectiveness over *all* hyperparameters (on the other hand, Fig. 5 computes the regret of the top-performing hyperparameters).
  - We included a guide for how practitioners should interpret all of these hyperparameter transfer results (Appendix Section C.3).

- We combined our existing algorithm and statistics experiments to give deeper experimental insights into the relationships between properties of the search space and the performance of different algorithms. The strongest correlations were as follows:
  - Regularized evolution performs comparatively much better on search spaces with small neighborhood sizes.
  - Local search performs comparatively better on small search spaces and also search spaces with small neighborhood sizes.
  - When tuned, RF and XGBoost perform comparatively much better on large search spaces and also search spaces with large neighborhood sizes.
  - We also showed there is a strict negative correlation between search space sizes and interquartile range, for NAS-Bench-101 and NAS-Bench-201.
- We added new tutorial notebooks for 1) getting started with the codebase, and 2) understanding the search space interface in NAS-Bench-Suite. We also changed the file structure of the runner files and scripts to follow best coding practices.

- We made other smaller fixes such as improving figure captions, making certain paragraphs clearer, and adding error bars to Fig. 3.

If you have any further comments or concerns, just let us know. Thank you for your time!

---

### Decision · Program_Chairs · 2022-01-20

**Decision:**

Accept (Poster)

**Comment:**

This paper proposes a novel benchmark for neural architecture search methods, which consists of 25 different combinations of search spaces and datasets. The main motivation is that existing NAS benchmarks, such as NAS-Bench-201, consider very small search space and few datasets, such that conclusions drawn with them do not generalize to unseen settings with different search spaces and datasets. The authors first describe the 25 different combinations of the search space and tasks for the given benchmark, and then conduct an extensive empirical study of existing NAS methods and performance predictors with the proposed benchmark, to show that architectures and hyperparameters found with the popular benchmarks do not generalize to other settings, which is consistent with their assumption.

—

All reviewers were initially positive about the paper, and remained positive throughout the discussion period. The reviewers found the paper well-motivated, and the proposed benchmark useful, as they agree with the need of introducing a single, unified framework that can validate a NAS method under diverse settings, since existing benchmarks only consider specific datasets and search spaces. However, the reviewers were also concerned with the weak technical novelty (Reviewer 2xvD), and that the work lacks deeper insights that could guide the community towards better methods (Reviewer Gku7).

I also agree with the authors and the reviewers on the necessity of having a unified benchmark that incorporates all different settings considered in the previous benchmarks, and find the extensive empirical study of existing NAS methods useful.

However, I find the work as rather technically weak as mentioned by R2xvD, since the authors spent too much time describing and showing the limitations of existing benchmark methods, while what is more important for benchmarks, is to justify how the proposed benchmark can evaluate the performance of different methods in a fair manner, while being representative of the practical settings. In short, the authors need to justify their design choices. Yet, the 25 settings proposed in the paper seem to have been arbitrarily chosen, and it is not clear if having a good performance on this benchmark is indeed a fair evaluation, or well-reflects how the NAS method will perform in practice. The proposed benchmark also does not really consider a novel search space or setting that have been overlooked in the past either, and does not provide much insights on the problem, as mentioned by Reviewer Gku7.

Thus, although I recommend an acceptance for its practical value acknowledged by the reviewers, the authors need to put a considerable amount of effort in revising the paper, and If this were a journal submission, the paper may need to undergo a major revision. Most importantly, as described, the authors should justify their design choices as well as whether evaluating a model on the benchmark yields “fair” and “representative” results, focusing more on describing the proposed benchmark itself.

---

> ### Public Comment · ~Colin_White2 · 2022-01-29
> **Thank you**
>
> We thank all reviewers and the meta-reviewer for their service, and we are delighted to see that our work is accepted to ICLR’22, based on the motivation, usefulness, and need for a unified framework to validate NAS in diverse settings. We are following the request of the meta-reviewer to revise our paper before the camera-ready deadline, focusing on justifying our design choices. To make it easier for future readers who might see this thread, we will also give answers to the questions here.
>
> > In short, the authors need to justify their design choices.
>
> We thank the meta-reviewer for bringing this up, because we have a simple answer that we will make clear in the camera ready version: we chose this set of 25 settings by using (nearly) *every* publicly available tabular/surrogate NAS benchmark.
>
> Our goal with NAS-Bench-Suite is to stay up to date by including *every* queryable NAS benchmark. Since the time of submission, we have added two more: NATS-Bench-SSS, and Surr-NAS-Bench-FBNet.
>
> > I find the work as rather technically weak … [the work] does not really consider a novel search space or setting
>
> Technical novelty was indeed required to create a unified interface to all the 25 queryable NAS benchmarks, due to their very different search spaces. We believe that it would be suboptimal to conflate the collection of existing benchmarks with the creation of a new benchmark or a new method in the same paper – indeed, that could be perceived as creating a bias in itself.
>
> > too much time describing and showing the limitations of existing benchmark methods
>
> We note that these limitations justify the need for NAS-Bench-Suite. Our findings in Section 4 and Appendix C also can be beneficial for future researchers, independent of NAS-Bench-Suite.
>
> > justify … whether evaluating a model on the benchmark yields “fair” and “representative” results
>
> We thank the meta-reviewer for bringing up this important point. Since we added nearly every queryable NAS benchmark, our work is “fair” in the sense that it is non-discriminative. Practitioners using NAS-Bench-Suite have the option to choose their own hand-picked subset of the 25+ tasks based on their specific application. We indeed agree that, in the future, it would be useful to explore this direction further and define representative subsets of the growing number of benchmarks in NAS-Bench-Suite based on application type. We will add this to the future work section of the revised paper.
>
> We once again thank the meta-reviewer for their valuable comments. We are happy to continue this conversation (and now that it is public, with anybody who has more questions or comments).